# Video-rate multi-color structured illumination microscopy with simultaneous real-time reconstruction

Andreas Markwirth [1], Mario Lachetta[1], Viola Mönkemöller [1,2], Rainer Heintzmann [3,4], Wolfgang Hübner [1], Thomas Huser [1] & Marcel Müller [1,2]

Super-resolved structured illumination microscopy (SR-SIM) is among the fastest fluorescence microscopy techniques capable of surpassing the optical diffraction limit. Current custom-build instruments are able to deliver two-fold resolution enhancement with high acquisition speed. SR-SIM is usually a two-step process, with raw-data acquisition and subsequent, time-consuming post-processing for image reconstruction. In contrast, wide-field and (multi-spot) confocal techniques produce high-resolution images instantly. Such immediacy is also possible with SR-SIM, by tight integration of a video-rate capable SIM with fast reconstruction software. Here we present instant SR-SIM by VIGOR (Video-rate Immediate GPU-accelerated Open-Source Reconstruction). We demonstrate multi-color SR-SIM at video frame-rates, with less than 250 ms delay between measurement and reconstructed image display. This is achieved by modifying and extending high-speed SR-SIM image acquisition with a new, GPU-enhanced, network-enabled image-reconstruction software. We demonstrate high-speed surveying of biological samples in multiple colors and live imaging of moving mitochondria as an example of intracellular dynamics.

[1] Biomolecular Photonics, Faculty of Physics, Bielefeld University, Bielefeld, Germany. [2] Laboratory for NanoBiology, Department of Chemistry, KU Leuven, Leuven, Belgium. [3] Leibniz Institute of Photonic Technology, Jena, Germany. [4] Institute of Physical Chemistry and Abbe Center of Photonics, Friedrich-Schiller-University Jena, Jena, Germany. Correspondence and requests for materials should be addressed to T.H. (email: thomas.huser@physik.uni-bielefeld.de) or to M.M. (email: muellerphysics@gmail.com)

Structured illumination microscopy (SIM), in its most widely used implementation as introduced by Gustafsson[1] and Heintzmann[2], enhances resolution by extracting high spatial frequency information from multiple fluorescence images, each acquired while the sample was illuminated with a different, defined sinusoidal illumination pattern. The illumination pattern is generated by interfering two coherent excitation beams in the focal plane of the objective lens, and this version of the SIM process, today often denoted as "coherent SIM", enhances the lateral resolution up to twofold compared with a wide-field image.

By adding a third beam, the technique can be extended toward three-dimensional resolution improvement[3], and today all commercially available SIM microscope platforms employ the three-beam SIM method and image reconstruction process. The availability of these instruments, their high imaging speed and the wide compatibility of SIM with existing fluorescent dyes and staining protocols have made it a very popular microscopy technique[4–9], especially for super-resolution live-cell imaging. Protocols detailing the SIM data acquisition and reconstruction process[10] enable efficient use of the technique.

Two-beam SIM systems operating at high speeds are also being widely used. These setups are arguably easier to design and more cost-efficient to build. Also, higher resolution implementations employing total internal reflection fluorescence excitation (TIRF) naturally require omitting a central beam. With the advent of objectives with ultra-high numerical apertures, and combined with non-linear processes, users of these systems were able to push the resolution improvement even further[11–14].

Many biological structures, however, cannot be imaged in TIRF mode, as their relevant features are not close enough to the glass coverslip. For this case, recent work on optical transfer function attenuation demonstrated that, by trading some of the gain in lateral resolution, optical sectioning can be achieved in the 2D reconstruction of two-beam SIM data[15–17]. Since the number of interfering beams directly determines the number of raw images required for near-isotropic 2D image reconstruction (9 for two-beam SIM, 15 for three-beam SIM), two-beam SIM runs 1.66× faster than three-beam SIM at equal raw image acquisition speeds. This makes it a very promising approach for video-rate live-cell imaging.

We recently introduced a new, open-source reconstruction software capable of reconstructing images obtained by all commercial and custom-built SIM platforms[18]. At the same time, Lu-Walther et al. provided a versatile blue-print for a robust, video-rate capable, spatial light modulator (SLM)-based two-beam SIM system, denoted "fastSIM"[19]. Combining both approaches readily allows research groups to build a state-of-the-art SIM platform from scratch. If, however, SIM were to resemble the handling and workflow of widely deployed wide-field and confocal fluorescence microscopes, it would become even more suitable for and broadly applied in biological research.

Currently, the use of SIM microscopes, both home-built and commercial, separates the image acquisition and reconstruction process, where the high-resolution image only becomes available after a dedicated post-processing step in the workflow. In our own strives for super-resolution live-cell imaging[20,21], this proved to be a significant drawback. For example, imaging of liver sinusoidal endothelial cells, the main features of interest, the cellular fenestrations, cannot be resolved by wide-field fluorescence microscopy, which is commonly used to first navigate the biological specimen to identify features promising for super-resolution imaging. Thus, time-consuming back-and-forth switching between measurement and post-processing steps is required just to find an initial area of interest.

This inspired us to recreate, modify and extend both the fastSIM approach and our image reconstruction software. The result is a 2D SIM setup allowing for simultaneous imaging with at least two excitation wavelengths at video rate (25 reconstructed frames per second or more), while providing immediate, real-time SIM reconstructions with sub-second time delay between measurement and displaying the resulting images. We demonstrate the performance of this system by navigating biological specimen at high-speed in the search for sites of interest, by high-speed imaging of diffusing microspheres, as well as high-speed imaging of intracellular dynamics—all while instantaneously displaying image reconstruction results to the experimentalist.

## Results

**Video-rate acquisition and immediate reconstruction.** While our SIM platform provides all the same essential features of a high-speed, coherent SIM microscope based on electro-optical devices (see Fig. 1a), its ability to instantly reconstruct image data acquired in multiple different color channels permits entirely new imaging work flows which are currently not possible with any other super-resolution microscopy method.

The performance of our approach is accomplished by optimizing both the acquisition and data processing speed of SIM imaging, and by interleaving both into a unified process. We utilize up to three dedicated sCMOS cameras to image each color channel individually. The use of separate cameras for the different color channels permits the realization of faster imaging modes, because addressing these cameras in an interleaved timing scheme (see Fig. 2a) provides the fastest possible performance. Each camera is connected to a dedicated personal computer (PC) by high-speed dual camera-link connectors for raw data transfer and camera control (see Fig. 1b). The camera computers are connected to a dedicated image reconstruction computer by two Gigabit Ethernet lines, permitting the rapid transfer of camera data for near-instantaneous processing on a graphics processing unit (GPU, see Fig. 1c). All components are triggered precisely by a low-cost microcontroller and a control PC, through which all imaging modes can be seamlessly selected and set (see Fig. 1b, Supplementary Figs. 4–6).

For each color channel, the system acquires nine raw SIM images, consisting of three phases and three rotations of the SIM pattern per time point. This scheme allows for a direct mathematical reconstruction of the super-resolved image[1,3,22] which can be achieved by a robust and efficient reconstruction algorithm. Standard background reduction algorithms for 2D SIM can be applied on the fly[16,17] and all raw data are stored to still allow for more advanced and especially computationally expensive post-processing[23,24]. An in-depth description of the full opto-mechanic and electronic design of the system is provided in the Methods section and Supplementary Note 1. The tight integration of the time-critical image acquisition process with rapid, on-the-fly image reconstruction and display, essentially provides this SIM platform with the look and feel of a standard wide-field fluorescence microscope, permitting a user to choose and focus on sites of interest, to acquire multicolor image data at rates well beyond video rate, and to share this experience with other users during the imaging process—all with optical super-resolution.

To demonstrate the on-the-fly image reconstruction of a biological specimen, we first imaged U2OS osteosarcoma cells, which were fluorescently stained to highlight the active mitochondrial membrane (MitoTracker Green, Thermo Fisher Scientific) and the endoplasmic reticulum (ER-Tracker, Thermo Fisher Scientific) (see Fig. 3). Figure 3a shows images taken from Supplementary Movie 1 at different time points, which display the screen of the computer that is running the GPU-enhanced SIM reconstruction software. The window on the right hand side

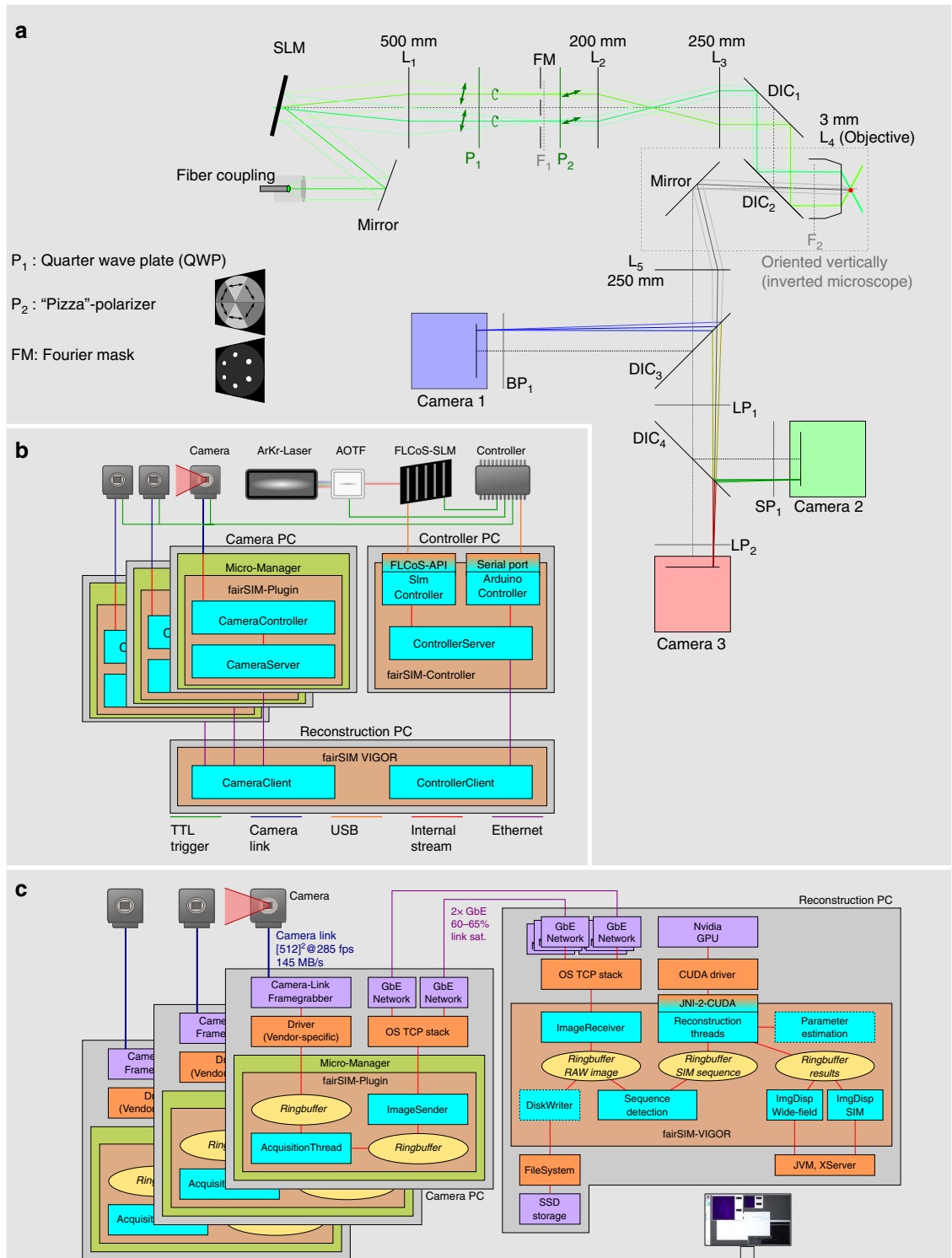

**Fig. 1** Schematic overview of the setup. **a** Opto-mechanics are based on ref. [19], modified for achromatic excitation and multicolor detection, for details see Supplementary Fig. 8. **b** The control flow diagram shows system components receiving both real-time timing pulses (TTL) and network-based control commands, for details see Supplementary Fig. 4. **c** The data flow diagram details how raw data acquired by the cameras travels through the on-the-fly reconstruction pipeline, being cached in ring-buffers for robustness and optionally saved for later, offline analysis. See Supplementary Fig. 3 for more details

of the computer screen shows the wide-field fluorescence images of the sample, while the window on the left hand side shows the on-the-fly reconstructed SIM image of the same sample location. The remaining columns of Fig. 3 show the offline reconstructed image data of the two color channels. When the sample is moved or the focus is adjusted, the wide-field view reacts immediately

and the reconstructed SIM images are displayed with minimal time lag of at most 250 ms, as can be seen in Supplementary Movie 1. The immediately available super-resolved images were used to check the cells for specific regions of interest.

High-speed, instant super-resolved frame rates can be achieved, which is demonstrated in Fig. 4. Here, freely diffusing

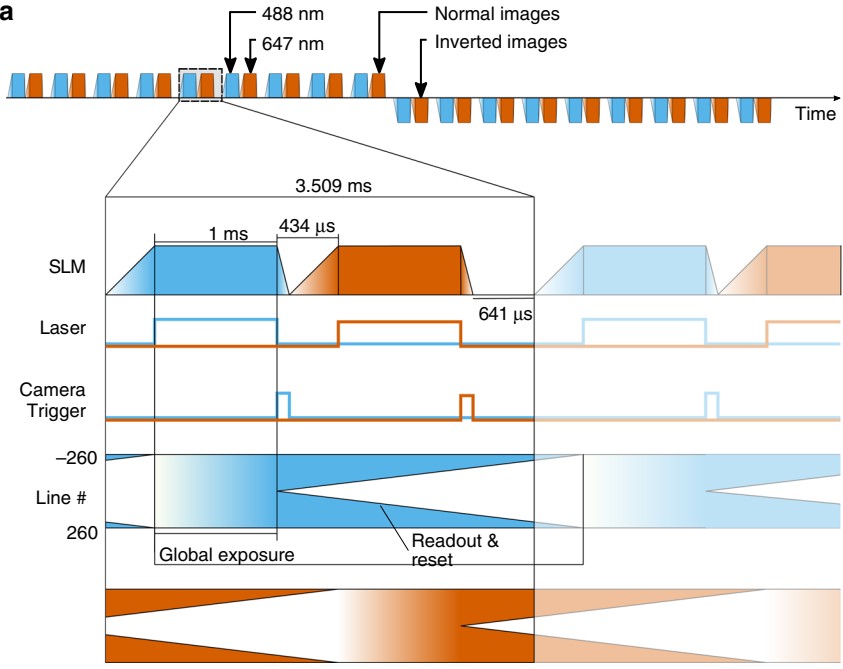

Total camera exposure time is ~3.5 ms: 1 ms + (260 lines × 9.65 µs/line)

**b**

| Region of interest (ROI) | No. of color channels | Exposure time ms/raw frame | Duty cycle single/multi color | Max SIM frames/s |
|---|---|---|---|---|
| 512 × 512 pixels (40 µm)$^2$ | 1 | 5 | 0.46 | 1 × 10 |
| | 2 | 2 | 0.41/0.82 | 2 × 22 |
| | 2 | 1 | 0.28/0.56 | 2 × 31 |
| | 3 | 1 | 0.23/0.69 | 3 × 25 |
| 256 × 256 pixels (20 µm)$^2$ | 1 | 0.5 | 0.28 | 1 × 63 |
| | 2 | 0.5 | 0.26/0.53 | 2 × 59 |
| | 3 | 0.5 | 0.17/0.53 | 3 × 39 |

**Fig. 2** Timing diagram and overview of achievable frame rates. **a** The diagram shows the intricate timing of all components (SLM, cameras, AOTF/Laser) during a high-speed SIM acquisition for two colors, with 1 ms exposure time and a 512 × 512 pixel ROI. The cameras have to be set to 520 × 520, as explained in Supplementary Note 1. Illuminating the sample in one color channel while the camera of the other color channel is reading out its sensor gives a speedup over an image-splitter approach with only one camera. More detailed diagrams are shown in Supplementary Fig. 1 (two-color) and Supplementary Fig. 2 (three-color). **b** The table shows the highest achievable frame rates for a selection of different combinations of imaging area, number of simultaneously imaged color channels and exposure times (rounded down to integers)

fluorescent microspheres were imaged with 0.5 ms exposure time per raw frame at 57.8 reconstructed SIM frames (520 raw frames) per second. The field-of-view is spanning 20 µm × 20 µm (256 × 256 pixels), which can be expanded to 40 µm × 40 µm depending on speed requirements (see the table in Fig. 2b). It should be pointed out, that the sample dynamics are readily observed in real time and super-resolution during the experiment. The short imaging time leads to low motion blur, even with such quickly moving objects, while the high frame rate results in a high temporal resolution. Figure 4b displays the rapid sample diffusion as a kymograph spanning a total of 1.85 s or 107 super-resolved images, respectively. A reconstruction scheme using a sliding window over subsets of raw frames (as demonstrated in ref. [25]) could of course also be applied to these datasets if even smoother reconstructed video is required.

**Live-cell compatibility**. We then employed the system for an established live-cell imaging experiment for high-speed super-resolution SIM[6], the study of organelle dynamics[26]. We stained

mitochondria in U2OS cells with MitoTracker green (Thermo Fisher Scientific), a widely used, commercially available, live-cell compatible organic dye. The ultrastructure of mitochondria is of great interest for unveiling its relationship between structural changes and mitochondrial function, but cannot be resolved in conventional wide-field imaging. We were able to image dynamics with sub-second frame rates and 5 ms individual frame exposure times, thus avoiding motion blur, over a prolonged time. Figure 5 shows excerpts from this time-lapse acquisition and highlights the dynamics, Supplementary Movie 2 provides the full, reconstructed dataset. The ultrastructure (mitochondrial membrane, cristae) can clearly be observed in the SIM result, while it does not become apparent in the wide-field images (see especially Fig. 5b). Concerning imaging speed, we found that, while our microscope system would have been able to image at even higher frame rates, the efficiency of the fluorophores, combined with the available laser power, limited us to 5 ms exposure time (see Fig. 2b). Even higher laser powers and thus higher imaging speeds would severely limit available observation time due to photobleaching. In addition, a second live-cell

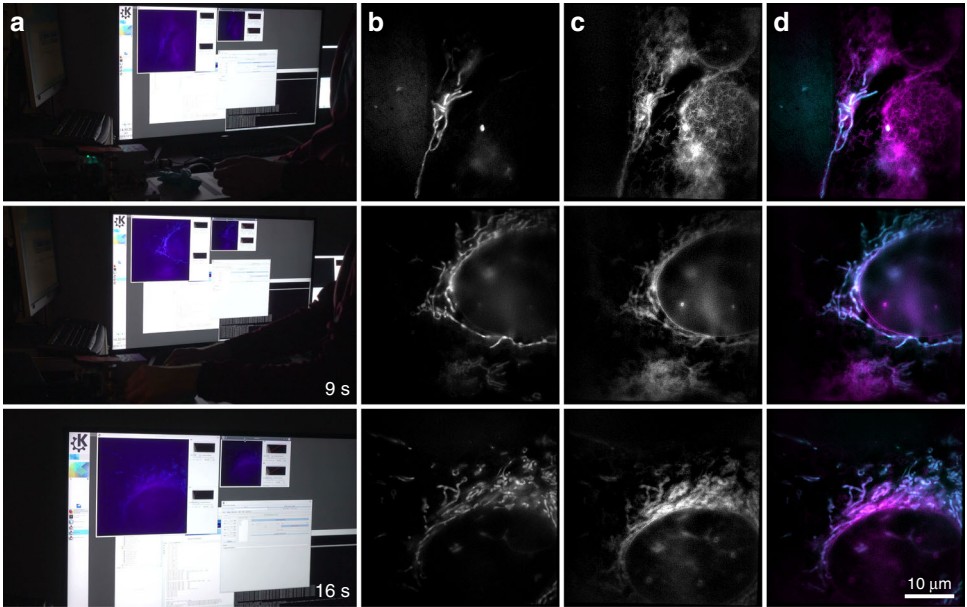

**Fig. 3** Demonstration of high-speed live-cell imaging and instant image reconstruction. A typical survey of stained U2OS cells was carried out with our SIM system. Video-recording (full version in Supplementary Movie 1) of the real-time results as presented to the user (column **a**), and the SIM reconstructions of the 488 nm channel (column **b**), stained for mitochondria (MitoTracker Green), and of the 647 nm channel (column **c**), stained for the endoplasmic reticulum (ER-Tracker Red). Column **d** shows an overlay of both reconstructed channels. Imaging was performed in dual color at 10.4 fps with 2 ms illumination time per raw frame (see Fig. 2b), multiple cells were surveyed in quick succession (see timestamp in each row). SIM allows to clearly resolve mitochondria and the endoplasmic reticulum

experiment was carried out to demonstrate three-color imaging of living cells. U2OS cells were stained with SYTO 9, MitoTracker Red, and Tubulin Tracker Deep Red (Thermo Fisher Scientific). This experiment was conducted in a time-lapsed mode to minimize photobleaching using 10 ms illumination time per raw frame and a frame rate of 2.4 SIM frames per second. The results are shown in Fig. 6 and Supplementary Movie 3.

Again, it should be pointed out that the SIM reconstructions, including the mitochondrial dynamics (Fig. 5, Supplementary Movie 2), were observed by the microscope operators in real time during the experiments. This allowed for direct interaction with the experiment, i.e., to move between regions of interest, note down events of particular interest, and discard experiments not of interest, without having to wait for any post-image-processing. Manual offline analysis is only needed if fine-tuning (filter parameters, cut-offs) is desired on specific datasets, as with any other imaging technique. This transforms the imaging workflow from a two-step process with limited feedback during data acquisition to an integrated approach, where full super-resolution data are immediately available to the microscope operator.

Besides providing immediately reconstructed SIM images, the system is capable of recording the acquired raw data stream at full imaging speed. Thus, finer filter adjustments and more time-consuming post-processing is possible at a later, offline stage. To demonstrate this, SIM data of the living U2OS cells shown were also processed by the published Hessian denoising matrix procedure, resulting in efficient noise and artefact removal (see Fig. 5)[24].

## Discussion

We have demonstrated a multicolor structured illumination microscope capable of video-rate imaging. The system displays an immediately reconstructed SIM image to the user, which allows it to operate much closer to how a conventional wide-field or confocal system is used. All data post-processing is performed in real time and automatically. This allows for immediate visualization of the super-resolved SIM images, and thus to quickly survey samples or to adjust the experiment conditions on-the-fly.

In our case, the system was used to survey the endoplasmic reticulum and the ultrastructure of mitochondria in U2OS osteosarcoma cells, which cannot be resolved in wide-field microscopy. The microscope enables the fast survey of large areas for cells of interest. As the sample stage is moved laterally and the focus is being adjusted, immediate feedback of a wide-field and super-resolved SIM image is being provided to the microscope operator. The super-resolved image will clearly show if sample features are in focus, which can be difficult to judge if only the wide-field view is available. Image contrast and zoom level can be adjusted in real time to visualize features of interest best and navigate the sample (see Supplementary Movie 1). All raw data are also stored on request to allow for flexible post-processing, e.g., enhanced denoising steps not currently realized in real time. This integrated workflow provides a significant improvement compared with the classical approach of separated imaging and post-processing steps. It should also be noted that the same performance of instant image reconstruction can also be achieved during post-processing with the stored raw data, which allows for the rapid optimization of image reconstruction parameters with instant feedback on their effect, which significantly speeds up any post-image-processing steps.

The high achievable frame rate was also used to study fast movements of organelles in living cells, by imaging mitochondria in U2OS cells. SIM super-resolution allowed us to resolve the mitochondrial ultrastructure, and the dynamics could be observed on a sub-second timescale.

In addition, the system enabled us to directly image freely diffusing microspheres and to track their movement with essentially no motion blur. The instant image reconstruction capability of the SR-SIM system enabled us to directly follow these processes in real time during the execution of the experiment, thereby greatly simplifying the identification of fast moving structures.

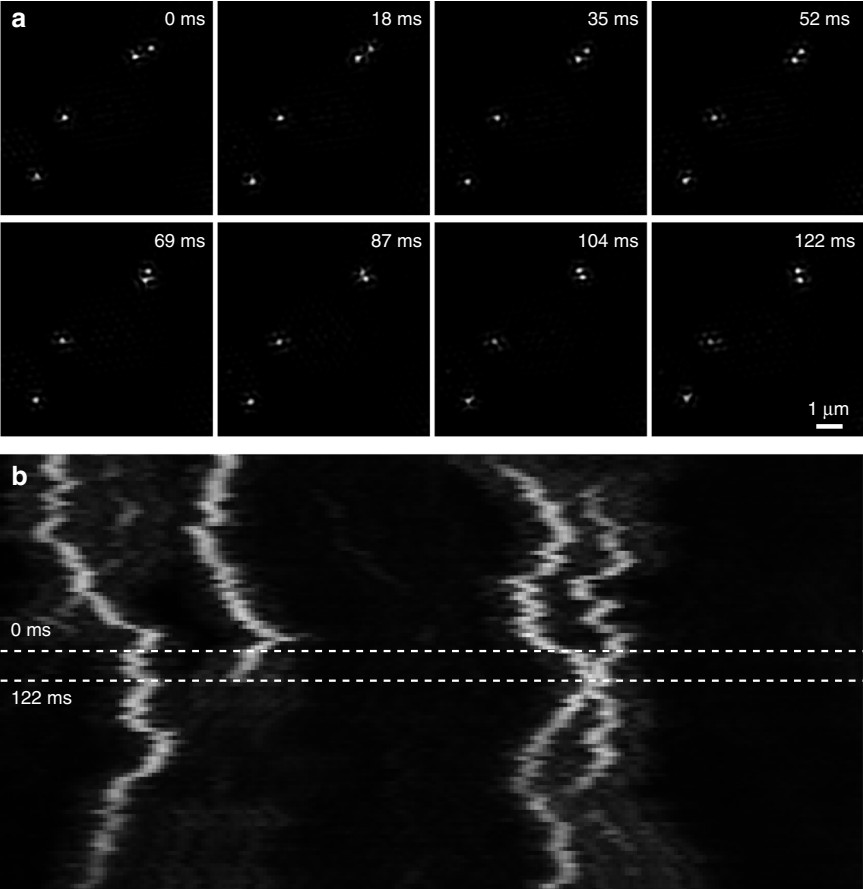

**Fig. 4** Demonstration of the high image frame rates achieved by high-speed SR-SIM. **a** Cutouts of eight consecutive SR-SIM frames taken at 57.8 SIM frames (520 raw frames) per second, with 0.5 ms illumination time for each of the nine raw frames per SR-SIM image. The sample consists of 0.2 μm TetraSpeck Microspheres in a 50/50 mixture of stock solution and glycerol. The excitation wavelength was 647 nm, laser power was about 3 mW before entering the objective lens. The short acquisition time allows the acquisition of images with low motion blur, even though the sample is diffusing quickly, and the high SR-SIM frame rate leads to a high temporal resolution. **b** Movement of the diffusing microspheres along the x-axis over time displayed as a kymograph. It spans over 107 consecutive SIM frames, or ~1.85 s, including the images shown in (**a**). Note that the diffusing microspheres could also be observed in real time and super-resolution by the system operator. Fixed microspheres were also imaged to demonstrate the general capabilities of the SIM system, see Supplementary Fig. 7

Such capabilities will find applications, e.g., in biosensing, following structural changes in macromolecular complexes, or signaling events in cells, to mention just a few.

It should be noted that multifocal image scanning microscopy techniques[27–29] or iSIM[30,31] also achieve very high image rates of up to 100 Hz. They complement coherent SIM well, as they offer a different tradeoff between imaging robustness in thick, scattering samples and lateral super-resolution. They do, however, also require a time-consuming post-processing step employing deconvolution in order to achieve their ultimate resolution improvement, so they could benefit from the methods described in this paper.

Other complementing super-resolution techniques include those based not on excitation light modulation, but on analyzing the stochastic fluctuation or blinking of fluorescent probes. Both, single-molecule localization techniques, which rely on very high fluorophore sparseness[32,33] and stochastic techniques, which are able to cope with much higher emitter densities[34,35], already feature implementations of accelerated, on-the-fly data processing, for example[36] for single-molecule localization microscopy (SMLM) and[37,38] for super-resolution radial fluctuations (SRRF) microscopy. Arguably, these techniques are somewhat orthogonal to SIM: They require much less complexity in imaging instrumentation, with data acquisition typically possible on a high-end wide-field microscope equipped with laser light sources and a fast, sensitive camera. However, they require specific fluorophores and sample conditions to work well, and typically need to acquire some hundreds (SOFI, SRRF) to thousands (dSTORM) of images for a super-resolved result.

The system described here provides flexible imaging modes, allowing the operator to rapidly switch between illumination times and inter-frame delays (for time-lapse imaging), which permits the fast adjustment of imaging conditions. In live-cell experiments this is particularly useful, as the currently available fluorescent dyes are rather prone to photodamage. For example, one could envision to first navigate the sample in a low damage mode, i.e., by capturing one image per second with short exposure time to avoid motion blur, and then switch to the full frame-rate high-speed mode once a region of interest exhibiting interesting sample dynamics is found. With instant image processing as demonstrated, here, this could even be automated to enable a new form of intelligent microscopy. Data can be recorded continuously, or recording can be limited to these observations of interest. All image frames are automatically timestamped with absolute dates and high-resolution time-stamps for ease of later analysis. Sample metadata (for example, cell type, dyes, excitation wavelength) can also be added to the data files. Furthermore, all raw data are retained for offline post-processing, so more

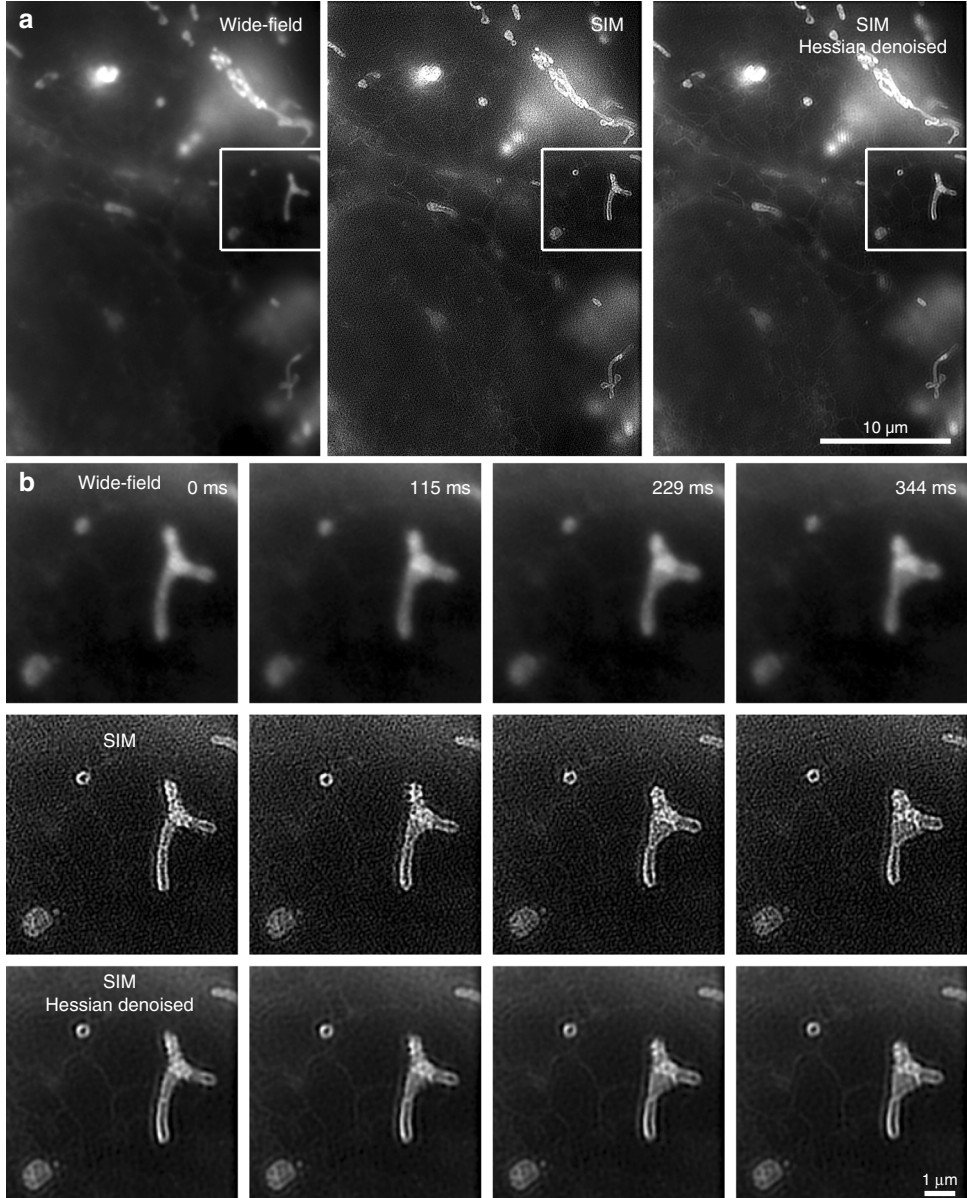

**Fig. 5** Demonstration of the system in use for live-cell imaging of mitochondrial dynamics. Single-color imaging of U2OS cells with 5 ms raw frame exposure time (see Fig. 2b), with overview (**a**) and time series of cut-out (**b**). The dynamics could be observed in real time and super-resolution by the system operator. The microscope hardware could easily provide higher speeds and continuous data acquisition, however, current dyes are limited in both brightness and photo-stability. Nonetheless, short exposure times are still very helpful in time-lapsed acquisition, as they suppress motion blur[46]

time-consuming processing algorithms (such as Hessian SIM) can be utilized to image with reduced excitation power and photodamage.

The opto-mechanical fastSIM design, our multicolor extensions, control components (electronics and software) and of course the GPU-accelerated on-the-fly reconstruction software are freely available to the community as part of the fairsim.org project and under open-source license (GPLv2 and later). With a modular layout, this should allow both to recreate and adapt our approach, and to add on-the-fly reconstruction to existing SIM microscopy systems. Furthermore, alternative image reconstruction methods, e.g., the sliding window processing of SIM image data, can easily be incorporated into the VIGOR platform and will then permit SIM imaging frames rates at the level of full raw image frame rates (e.g., at 0.5 ms exposure time), albeit resulting in averaging of the sample dynamics over at least nine raw image frames[25].

The SIM imaging modality was optimized for speed (three angles, three phases) and the pattern was chosen to give a lateral resolution improvement of 1.8 (at an NA of 1.33). This leaves some overlap of the side bands and thus allows for some out-of-focus reduction by reweighting the contribution of the different bands[16]. Two trade-offs are possible for optimization, both of which only require minimal changes to the microscope (replacing the order-selection-aperture and realigning the polarization filter): A coarser pattern set could be used to increase the overlap. This would improve the background suppression without the need to record more images but reduce lateral resolution improvement. Alternatively, a coarse and a fine pattern could be used consecutively. This would require additional images to be acquired and would thus slow down the instrument but allow for background reduction and full lateral resolution. Of course, three-beam SIM is preferable, here, but this also comes with higher effort for polarization control and

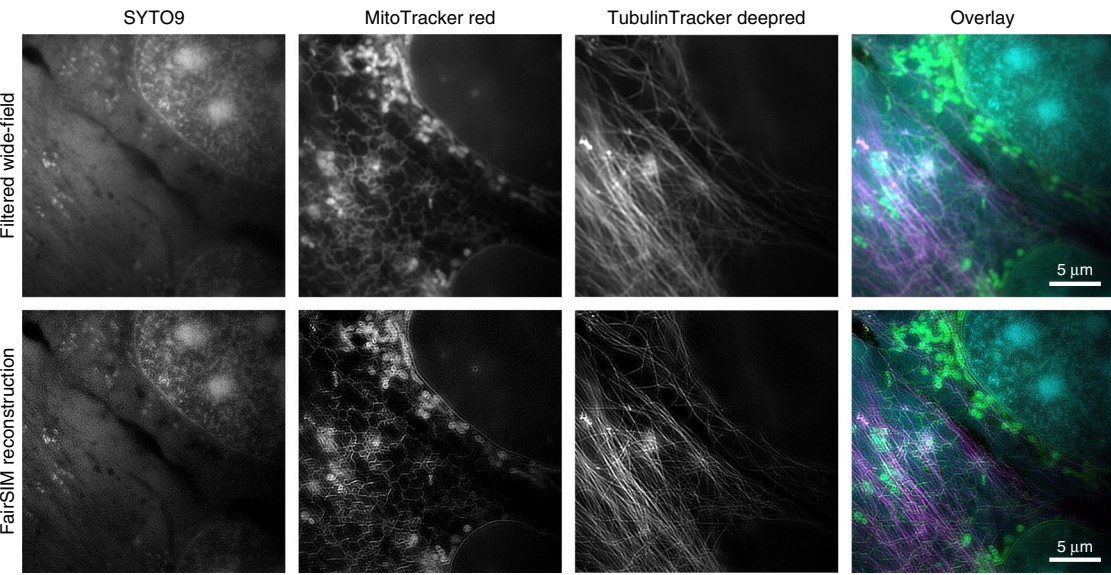

**Fig. 6** Demonstration of three-color live-cell imaging. These images were acquired with 10 ms raw frame exposure time and 2.4 (time lapsed) multicolor SIM frames per second. Live U2OS cells were stained for nucleic acids with SYTO 9, mitochondria and endoplasmic reticulum with MitoTracker Red, and polymerized tubulin filaments with Tubulin Tracker Deep Red. The individual channels and an overlay of all three channels (SYTO 9 in cyan, MitoTracker Red in green, and Tubulin Tracker Deep Red in magenta) are shown as reconstructed Wiener filtered wide-field and FairSIM images, respectively. The scale bar in the overlay images represents 5 μm

top-phase alignment (matching the axial pattern repetition to the detection focal plane).

We see our system as an open platform that can be easily extended toward future SIM imaging modalities. Implementations of non-linear SIM[11–14], TIRF SIM with highest NA objective lenses[14,39], and multi-plane detection schemes[40–42] can easily be added to the platform. Multi-modal imaging[21], for example using SIM to select areas of interest, followed by single-molecule localization for highest resolution imaging, can be carried out on the system.

For live-cell imaging, a combination of advanced, photo-stable dyes and recent advances in SIM post-processing algorithms[24] will allow for new insights into high-speed dynamics, both taking place on sub-second time scales and requiring sub-diffraction resolutions.

## Methods

**Microscope system concept**. Our setup builds on, modifies and extends the blueprints of the fastSIM[19] approach. We also extended our fairSIM software[18] with GPU- and network-support, to be able to reconstruct the multicolor SIM data at video rates with negligible delay so it can be immediately displayed to the user together with the wide-field data. By interleaving the acquisition of images in the first and second color channels, two-color data with 1 ms illumination time can be acquired at the same rate as one-color data, and a third-color channel only slows the acquisition speed down from 31 to 25 fps. In the following, we provide an overview of all major components.

**Laser light sources**. The excitation light source is an Argon-Krypton-Laser (Coherent Innova 70C Spectrum), set up to run in multi-line mode, providing excitation light at 488 nm, 568 nm and 647 nm. An acousto-optical tunable filter (AOTF, AA Optoelectronic AOTFnC-VIS-TN) enables us to quickly toggle the transmission of each wavelength. Real-time control is achieved through TTL signals for each wavelength, which are fed into a programmable control box of the AOTF. The laser light is transmitted through a high-power single mode 3 μm polarization maintaining optical fiber (OZ Optics PMJ-A3HPM,3S-633-4/125-3AS-5-1) to both clean up the beam-profile and to allow the laser to be placed on a separate optical table. Originally a lens-based collimator (Linos Fok.-Koll. MB 06) was used, but it suffered from chromatic aberrations: The output beam could not be collimated for all three wavelengths at the same time. A recently installed reflective collimator (Thorlabs RC12FC-P01) produces much more promising results.

The combination of AOTF and Argon-Krypton-Laser was chosen due to availability in our laboratory. A set of diode lasers providing the desired

wavelengths at sufficient power levels should perform equally well, while requiring less day-to-day maintenance.

**Multicolor achromatic optical path**. The following optical path is similar to the fastSIM[19] approach, but switches out all filter components with achromatic components: A collimated laser beam with 12 mm diameter illuminates the spatial light modulator (ForthDD SXGA-3DM) under an angle of ~4°. The SLM is a binary phase-shift device: All its pixels always reflect light, but depending on their on/off state, rotate the polarization by an angle. While the parallel components of the polarization of the reflected beams are the same for pixels in both on- and off-states and form the zeroth order maximum, the anti-parallel components could just as well be described as having the same polarization, but a 180° phase-offset, resulting in an interference-pattern. Therefore, the SLM can be used as an electronically defined optical phase grating.

We use a made-to-order quarter-wave-plate (Thorlabs AQWP10M-450-650-SP with 22.1 mm free aperture) and an achromatic version of the fastSIMs custom-made segmented azimuthal polarizer (Codixx colorPol®VIS 038 BC3 CW01 25.4 mm). Finally, the excitation light is reflected onto the sample via two multi-band, imaging-quality beamsplitters (Semrock Di03-R405/488/561/635-t3-25x 36) and a 60×, 1.45 NA oil-immersion objective (Olympus PlanApo 60x/1.45 Oil TIRFM ∞/0,17).

The samples fluorescent emission passes the second multi-band beamsplitter, and is projected onto sCMOS cameras (PCO AG pco.edge 4.2) through a common tube lens. Two custom-made, imaging-quality dichromatic beamsplitters (AHF Analysetechnik 560 DCXR, AHF Analysetechnik 650 DCXRU) allow to use up to three camera ports: The first beamsplitter (560 nm longpass) separates out the 488 nm emission and the second (650 nm longpass) the 568 nm emission, while the 647 nm emission passes through both beamsplitters. Additional low- and band-pass filters are used to clean up the detection spectrum. Please see Supplementary Fig. 8 for more details.

The common tube lens was chosen due to space restrictions. A layout with separate tube lenses for each color channel and the dichromatic beamsplitters and filters placed in the parallel beam is preferred, as this would introduce fewer aberrations. We verified the performance of our arrangement by simulating this setup in optical design software and found that the proper arrangement of the second dichroic beamsplitter at 45° and rotated by 90° along the optical axis compared with the first beamsplitter, compensates astigmatism and coma to a large extent. In combination with the magnification provided by this setup we found this to have no detrimental effect on the imaging quality.

**SLM pattern generation and filtering**. The SIM pattern is generated by employing the SLM as a switchable, binary optical grating. Due to the binary pixel-structure of the device, its diffraction pattern contains spurious diffraction orders that are to be blocked from entering the objective. The relay lens system in the excitation path gives access to the SLMs Fourier plane and allows to filter out these unwanted orders with a simple mask.

For this approach to work, a pattern set has to be found where for each orientation of the pattern, all desired light transmits through the mask, while ideally all unwanted light is blocked. We extended the originally proposed pattern search algorithm[7,9,19] to generate optimized patterns for multiple colors. The position of the diffraction spots in the Fourier plane depends on the orientation and spacing of the pattern that is displayed on the SLM and on the wavelength of the light. The spacing of the pattern is adjusted for each color to generate the diffraction spots in the same position in the Fourier plane, so that the light can pass the stationary spatial filter mask. As a side-effect, this of course also ensures all wavelengths yield the same relative resolution improvement. We provide an implementation of the search algorithm as a user-friendly ImageJ plug-in.

Mechanically, the filter mask can be created by hand if necessary: A simple approach is to punch small holes through a thin film of aluminum, using a needle, and then carefully widening these holes if necessary. However, a more robust filter can also be created by manufacturing it from e.g., plastic or metal. The precision required (10 μm) is well within those of most workshops equipped for precision metal work. To obtain the precise Fourier spot positions, we placed a CMOS camera (UI-3060CP, IDS Imaging Development Systems) in the Fourier plane and ran the SIM sequence. To not damage or overexpose the camera, the central beam should be blocked and the laser power has to be greatly reduced (e.g., by neutral density filters in front of the fiber). The resulting image can be used for both creating a mechanical drawing of the filter mask, and also offers a good cross-check of the systems beam quality.

**Timing of SLM and camera**. The high image acquisition speeds of the fastSIM system can only be achieved by careful and exact synchronization between the spatial light modulator, the cameras and light sources. Please see Supplementary Note 1 for a description of the timing properties of both the ferro-electric SLM and the sCMOS cameras employed by the system. The multi-camera approach has been chosen over an approach with only one camera and a beamsplitter, as this enables the concurrent exposure of the sample and camera readout, which results in a speedup that is required to match the timing-constraints set by the FLCoS (see Supplementary Note 1 for details).

We found it useful to think of both the cameras and the SLM as devices alternating between two states, exposure and processing. During exposure the SLM displays a static pattern, and no readout is performed on the camera. During processing, the SLM switches pattern and the camera reads out. A synchronization scheme now has to ensure that both the SLM and one of the cameras are in exposure mode at the same time, and that each laser wavelength is only turned on while the modes overlap.

After each exposure, the SLM takes 0.44 ms to switch its pattern. Using only this time for camera readout would yield a field-of-view of only 80 lines, corresponding to 6.4 μm at a typical projected pixel size of 80 nm. For more reasonable fields of view (256 or 512 lines, i.e., 20 μm or 40 μm), the processing time of the camera exceeds that of the SLM, so the SLM sequence has to be delayed for 0.85 ms or 2.1 ms to wait for camera readout.

For single-color SIM (excitation at one wavelength), a rather elegant but somewhat complex approach[43] mitigates this effect by synchronizing a dissected set of SIM patterns with a line-triggered rolling shutter readout mode, at the cost of an overall longer sequence (14 partial patterns instead of 9).

For our setup, we deemed the use of multiple excitation wavelengths to be most practical for biological applications. Many samples will be stained for at least two structures, which should be imaged simultaneously. In this case, the camera readout time for one wavelength can be interleaved with the exposure of the other wavelengths. For example in the three-color, 1 ms mode, while the SLM displays a pattern for the 488 nm channel, the camera for that channel is in global exposure status, while the other two cameras are reading out their sensors.

A selection of possible combinations of color channels and exposure times, the resulting maximal achievable frame rate and duty-cycle can be found in Fig. 2b, while Supplementary Figs. 1 and 2 provide two examples of detailed timing diagrams for the 1 ms two-color and three-color modes.

An Arduino microcontroller (Arduino Uno) is used for the real-time synchronization. Due to timing-constraints of the FLCoS-SLM, which are described in Supplementary Note 1, no frame-by-frame synchronization is possible.

However, since the SLM timings are exactly known and very precise, this is not necessary: The microcontroller starts each sequence by triggering the SLM and duplicates the SLM timing sequence to trigger both laser wavelengths (via TTL to the AOTF) and camera readout (via TTL to the cameras) at the appropriate time points. A digital storage oscilloscope (LeCroy waveRunner 6100) was used to verify the timing sequences.

**Network transfer and reconstruction pipeline**. The reconstruction software in use by this project is based on our universal fairSIM[18] reconstruction plug-in. To achieve real-time reconstructions at video rate, with immediate display of the results, two large modifications and extensions of the software were necessary: The software had to be modified to allow for continuous reconstruction and display of the results, and it had to be sped up to the required processing-rate.

Continuous reconstruction is achieved by a multi-threaded, networked, and modularized framework. A first image acquisition module transmits raw data

frames from the camera computers to the reconstruction software, which can be run on a separate computer. Employing this network link, instead of running both image acquisition and reconstruction on the same computer, was chosen for flexibility: Existing systems (for example in commercial microscopes) only need to add a simple network transmit module to their camera computer, and do not need to be upgraded with GPU hardware. It also allows to freely mix different operating systems, and to scale up to multiple camera computers and reconstruction nodes. Finally, the networked approach keeps the camera computer free of workload from the SIM image reconstructions, so no frames are missed due to a highly busy CPU.

For camera control and readout, a MicroManager[44] plug-in is provided. We have tested this for compatibility with the pco.edge 4.2 and Hamamatsu Orca Flash 4 series cameras, but it should in general work with any camera system compatible with MicroManager. The software can bundle multiple network links on TCP/IP level, so inexpensive network hardware (in our case, two standard Gigabit Ethernet cards per camera computer) can be used. Additional details can be found in Fig. 1c and Supplementary Fig. 3.

During SIM image acquisition, raw image sets (nine images, covering three different angles with three different phases per angle) are extracted from the network input stream and passed to the reconstruction module. The reconstruction step provides a wide-field view and of course the high-resolution SIM image as an output, which are instantly displayed to the user, as shown in Fig. 3 and Supplementary Movie 1. In addition, the images can be registered (the determination of parameters for the image registration is based on bUnwarpJ[45], applying the registration matrix is based on integrated, performance-optimized code) and SIM reconstruction parameters can be re-estimated for each channel on user request. All these components are run in separate threads (allowing for speedup through multi-threading), communicate through ring-buffers (so short delays in the reconstruction pipeline do not cause frame drops) and are run in multiple instances (one for each color channel).

The estimated overall latency is only a few tenths of a second. In the two color, 1 ms mode, capturing the $2 \times 9$ raw frames takes about 32 ms. Transferring the last frame to the camera computer via Camera Link takes <1 ms, network transfer to the reconstruction computer via GbE takes an additional 4 ms. The registration takes ~10 ms, and adjusting the brightness, contrast and gamma takes ~30 ms ($1024 \times 1024$ pixels, two colors). Displaying the image can take up to 40 ms, possibly longer, depending on the monitor used.

On a mid-range consumer-level graphics card (Nvidia GTX 1060), the GPU-assisted (see below) SIM reconstruction step adds latency of about 20 ms to the image processing pipeline, including data transfer to the GPU and back.

**GPU-assisted SIM reconstruction**. Using a current multi-core desktop CPU, a 2D SIM reconstruction takes ~200 ms for a typical input image size of $512 \times 512$ pixels and output image size of $1024 \times 1024$ pixels. While this speed is sufficient for most post-processing applications, it is an order of magnitude too slow for the desired live reconstruction mode. Fortunately, the SIM reconstruction algorithm is very well suited for speedup through massive parallel processing, in our implementation by porting it to a CUDA-enabled graphics card.

The modular structure of our original software, together with the object-oriented approach of Java, allowed to tightly integrate a GPU-accelerated version with the original code. Basic, low level linear algebra functions (vector addition, scalar products and norms, point-wise multiplications) and Fourier transformations have been duplicated on the GPU. Through virtual functions, switches between CPU and GPU implementation happen automatically: If the data structures reside on the CPU, the Java implementation is used. If they reside on the GPU, the optimized and accelerated CUDA version is used.

This scheme ensures that high level code only needs to be written once. The same SIM reconstruction algorithm can thus run on the CPU (in portable, system-independent JAVA code) and on the GPU (in highly optimized CUDA code). Extensions and improvements to the reconstruction algorithm thus only have to be implemented once to be available both in a portable CPU and in a fast GPU version. For the live reconstruction mode, we also introduced two improvements to the GPU data transfer. One aspect is to reduce the PCI Express bus usage, which is achieved by directly copying the raw pixel data (2 bytes per pixel) instead of any floating-point numbers (4–8 bytes per pixel) to the GPU. The other improvement is to run the reconstruction of multiple frames concurrently. Thus, the GPU can copy over one dataset while still running the computation of the last one.

As a result, our setup, using a mid-range consumer-level graphics card (Nvidia GTX 1060), can reconstruct up to ~70 high-resolution frames per second (540 raw frames per second) with a resolution of $1024 \times 1024$ pixels (raw frames: $512 \times 512$ pixels), even though the reconstruction process, including data transfer, for a single SIM sequence takes about 20 ms.

The GPU enhancements of the SIM reconstruction process are of course not limited to the immediate reconstruction mode. Since they are integrated into the fairSIM project, they can also speedup the post-processing, which is especially helpful for the processing of time-lapse movies.

**System control software**. Our goal was that the fastSIM setup can be controlled via a single graphical user interface. For this purpose, interfaces between fairSIM-VIGOR (written in Java) and the separate devices of the setup (SLM, micro-controller & cameras) were developed and implemented. Since the individual

devices are attached to different computers, it was necessary to implement a network-based infrastructure for control communication between the computers (see Supplementary Figs. 3 and 4). The graphical user interface of the controller was integrated into the graphical user interface of fairSIM-VIGOR as so-called Advanced-GUI. This can be used to control the SLM, the Arduino and the cameras individually, which is most useful during initial system setup and alignment. In addition, the so-called Easy-GUI (see Supplementary Figs. 5 and 6) was developed, which enables the user to operate the setup in a simple and intuitive way. It is only necessary to select the desired excitation lasers and exposure times before the acquisition can be started. The Easy-GUI automates all required tasks, such as selecting SLM running orders, setting up cameras (exposure time, regions of-interest) and starting microcontroller programs, and thus hides the complexity of the system in day-to-day use.

**Limitations**. We found that rather than the laser power or the maximum frame rate of the microscope, it is often the brightness and stability of available fluorophores that limit the temporal resolution or duration of live-cell experiments: The total number of images that can be taken before the fluorophores bleach is similar to other SIM microscopes, so using a higher frame rate with the same number of detected photons per image (which of course relates to the image quality) limits the timespan over which the sample can be monitored. Our control software can introduce delays between SIM acquisitions, so that the user can adjust the frame rate without slowing down the SIM acquisitions. By doing this, one can take advantage of the short illumination times and rapid acquisition of raw frames, which reduces motion blur, while examining the sample with the frame rate that gives the best compromise between temporal resolution and bleaching for each experiment.

While time-lapse recordings of biological samples obtained with moderate laser powers to reduce bleaching or in the fastest modes (0.5 ms illumination time or less) are often quite dim and thus noisy in an (instant) reconstruction with fairSIM, they can still be processed offline with time-consuming algorithms that are more suited for low-SNR images, like Hessian SIM.

**Calibration samples**. Fluorescent beads, in our case 200 nm TetraSpeck Microspheres (0.2 μm TetraSpeck Microspheres, Thermo Fisher Scientific), have become a standard in structured illumination microscopy. Prepared to form a thin layer of clusters with closely packed beads, they offer a very distinct indicator for SIM super-resolution. In wide-field, single beads cannot be distinguished in the cluster. With the improved resolution through SIM, they can, however, be separated. These beads are used as calibration samples. After realigning the system, or when temperature changes introduce thermal drift, a bead sample allows to easily extract a new SIM reconstruction parameter set and image registration data. As the beads are very bright with little background, they also allow for a more precise estimate of the illumination pattern modulation depth than typical biological samples. This parameter is needed for the reconstruction process. Since the TetraSpeck beads emit light into all three color channels, a multicolor image can be used as input for bUnwarpJ to create the registration data.

For sample preparation, TetraSpeck Microspheres (0.2 μm, T-7280) were purchased from Thermo Fisher Scientific. Commercially available coverslips (~150 μm) with 24 × 60 mm in size were carefully cleaned with HellmanexIII (Hellma GmbH Göttingen, Germany) for 20 min in a supersonic bath at 45 °C. Then, the coverslips were rinsed two times with pure $H_2O$, followed by another 20 min in the supersonic bath in pure $H_2O$. Then, the coverslips were dried in an air flow prior use. A silicone sheet (self-adhesive; Sigma-Aldrich (GBL666182-5EA)) with a hole of 4 mm in diameter was disposed to the coverslip. A measure of 5 μl stock solution of TetraSpeck Microspheres was mixed with 5 μl $H_2O$ and vortexed for 2 min. The solution was dispensed onto the coverslip and dried headfirst over night at 4 °C.

**Diffusing microsphere sample preparation**. The TetraSpeck sample shown in Fig. 5 was prepared by dispensing 2 μl stock solution of 0.2 μm TetraSpeck Microspheres (Thermo Fisher Scientific) onto a clean #1 coverslip (Menzel-Gläser), adding 2 μl of Glycerol and mixing the two by repeated pipetting. The sample was prepared and imaged at room temperature.

**U2OS sample preparation**. U2OS cells (DSMZ, no. ACC 785) were seeded and grown on ibidi-chambers (ibidi, 80827) with #1.5 cover glasses in DMEM 10% fetal calf serum.

For Fig. 5, mitochondrial membranes and to some extent—as background—the endoplasmic reticulum were stained with MitoTracker Green (Thermo Fisher Scientific, #M7514) as described by the manufacturer. Endoplasmic reticulum and —as background—mitochondria were stained with ER-Tracker Red (Thermo Fisher Scientific, #E34250) according to the manufacturer.

For Fig. 6, U2OS cells were simultaneously stained with three fluorophores according to the manufacturer's instructions. Cells were incubated with SYTO 9 at 2 μM for 30 min (S34854, Thermo Fisher Scientific) in order to stain predominantly chromatic DNA with a distinct background signal in the cytoplasm as it has low affinity for RNA as well. MitoTracker Red CMXRos (M7512, Thermo Fisher Scientific) at a 500 nM concentration for 30 min stained the mitochondria as well as the endoplasmic reticulum. Finally, Tubulin Tracker Deep Red (T34077,

Thermo Fisher Scientific) stained the polymerized tubulin. Cells were washed three times with PBS after incubation with the respective staining solution and imaged in $CO_2$ independent growth media (18045054, Gibco, Thermo Fisher Scientific).

**0.04 μm FluoSpheres sample preparation**. 0.04 μm FluoSpheres Microspheres yellow-green (505/515) F8795, red-orange (565/580) F8794, and dark red (660/680) F8789 were purchased from Thermo Fisher Scientific.

One microliter of stock solution was vortexed, diluted in water (1–1000) and vortexed again. A small amount of this diluted solution was added to a #1.5 coverslip, smeared with the tip of the pipette and dried. The sample was mounted in VECTASHIELD (Vector Laboratories).

**Reporting summary**. Further information on research design is available in the Nature Research Reporting Summary linked to this article.

## Data availability

The data generated during and/or analyzed during the current study is available on Zenodo: https://doi.org/10.5281/zenodo.3295829.

## Code availability

All source code is openly accessible under GPLv2 (or later) license, and is accessible in repositories under https://www.github.com/fairSIM.

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

## Acknowledgements

This project has received funding from the European Union's Horizon 2020 research and innovation program under the Marie Skłodowska-Curie grant agreements No. 752080 (to M.M.) and 766181 (to T.H.). T.H. acknowledges funding by the Deutsche Forschungsgemeinschaft (DFG, German Science Foundation)—project number 415832635. R.H. acknowledges the support by the DFG through project B5 in the SFB/Transregio 166 and the Federal Ministry of Education and Research (BMBF) via FKZ 13N13140. We thank PCO AG for the loan of pco.edge 4.2 cameras for testing the multicolor imaging capabilities.

## Author contributions

A.M. constructed the opto-mechanics, control electronics and wrote parts of the paper. M.L. implemented the GUI and the network/control flow software V.M. performed the sample preparation and imaging R.H. provided the opto-mechanical design and details of the SIM image reconstruction process and helped with the project W.H. performed the sample preparation and imaging. T.H. conceived and supervised the research project. M. M. wrote the image reconstruction software, supervised and participated in the construction and implementation of the opto-mechanics and electronics of the microscope and its utilization during image acquisitions and wrote parts of the paper. All authors participated in the reading and editing of the final paper.

## Additional information

**Competing interests:** The authors declare no competing interests.

