## [Peer Review File · Nature Communications]

Reviewers' comments:

Reviewer #1 (Remarks to the Author):

In the paper Video-rate multi-color structured illumination microscopy with simultaneous real-time reconstruction by Andreas Markwirth and colleagues, the authors present a new approach, they dubbed VIGOR for open-source real-time reconstruction of SIM data using GPU-acceleration. The manuscripts aims to overcome the problem that acquisition and reconstruction are typically uncoupled in SIM imaging. The authors implement and adapt the earlier published fastSIM approach and combine it with a GPU accelerated network version of their fairSIM solution. The authors demonstrate their approach on mitochondria dynamics, beads, and fenestrations in LSECs. The article is timely and provides the reader with a blueprint to build a fast SIM system that provides an immediate result. This is an highly desired improvement for the workflow of SIM imaging. While the article is well and clear written, some questions and remarks remain and require a major revision.

The article is making an argument about using real-time reconstruction for super-resolution imaging. The authors should refer to the SRRF article (Gustafsson N, et al (2016). Fast live-cell conventional fluorophore nanoscopy with ImageJ through super-resolution radial fluctuations. Nature Communications, 7. doi: 10.1038/ncomms12471) as it offers super-resolution and has been implemented on ANDOR cameras directly for real-time processing.

It is understood that the authors not necessarily have an ANDOR SRRF setup or access to FPGAs or related chips on the camera to implement a similar scheme and it is appreciated that the solution presented here is openly available. However, the article should at least be discussed. Also since there is also an open source implementation a comparison of the resolution performance can be made.

The authors describe the possibility to use their implementation for life cell imaging and demonstrate this with biological samples. However, one concern is the photo-toxicity. Here the effective light dose should be discussed. It could be helpful for the reader to put the results it in perspective to other techniques like SOFI (Super-resolution optical fluctuation imaging, Dertinger and colleagues) related techniques; and SRRF, which promotes its applicability for live cell imaging and using conventional dyes like GFP. The advantages and disadvantages should be discussed. Again a direct comparison of data would be preferred.

In the article, the authors uses nine images for the reconstruction as compared to the higher precision 25 image reconstruction for obvious timing reasons. However, can a schematic for 25 images be deployed? There could be applications of slower dynamics combined with the need for more precision/resolution. How would the timing be affected?

Along the same lines but for speeding up the acquisition, Meiniel and colleagues, discuss schemes for REDUCING DATA ACQUISITION FOR FAST STRUCTURED ILLUMINATION MICROSCOPY USING COMPRESSED SENSING. Could these be used for the current implementation and could the imaging speed be still increased? Will these be implemented?

One of the biggest advantages of SIM is the ease to use multiple colors painlessly as compared to other super-resolution techniques. The article claims video-rate multi-color SIM imaging. In Figure 5 the use for live cell imaging using mitotracker is demonstrated. This figure is a missed opportunity and the potential to reconstruct a biological sample live in 3 colors should be demonstrated. Also proving and actually measuring dynamics is crucial here. Potential examples for applications would be focal adhesion points or the tracking of membrane receptors. While other options demonstrating the potential would be perfectly fine, the imaging of cell surface clusters offers the opportunity to compare the current approach and its potential with other higher resolution methods, like single molecule techniques, which are typically slower. What are the limits of the setup with respect to molecular quantities; can single molecules be imaged? Can single GFP molecules be visualized? In this context, it would be relevant for the reader to know to what extent the intensity values can be interpreted and be used for measurements.

It is appreciated that the authors time stamp the images and that additional metadata can be added. It would be desirable to automatically add the reconstruction parameters to distinguish different version later easily and likewise being compliant with standard OME (Open Microscopy Environment) formats for the metadata so that specific images can be searched on repositories.

One of the problems that haunt the reconstruction of SIM in my view is that the algorithm uses a noise estimate that depending on the sample can be off and needs to be adapted during the reconstruction. This holds true especially for dimmer samples, for reference see (Superresolution live imaging of plant cells using structured illumination microscopy, by George Komis and colleagues). The current manuscript demonstrates the live reconstruction of relatively bright samples. The question, therefore, is how to handle dimmer biological samples and how to adapt for example noise parameters or frequency weighting on the fly.

A cell can easily exceed the field of view described in the present manuscript; therefore the question is, can bigger areas be imaged?

Minor comments:

The authors use the camera link for their implementation. It is not clear if that is required or if the USB3 version would be sufficient, given that only 512X512 pixels are read out. Also what bit depth is used?

The authors use three computers for the three cameras. As they also implemented the Hamamatsu cameras and the HOKAWO software from Hamamatsu supports the parallel readout of multiple cameras on one computer. Could that be implemented similarly for VIGOR? What are the bottlenecks?

The authors use three cameras and if I understand correctly only read out 512x512 pixels. However, the chip size of the used cameras is 2048X2048 pixels. Somehow this raises the question if an 'old fashioned' image splitter that projects different wavelengths on the same chip can be used?

Could the timing of the cameras be used directly? It is understood that the timing depends on the SLM, but since the used cameras offer intricate trigger possibilities, it is intriguing to use a simpler glue logic, than an Arduino board. While the Arduino board is inexpensive, for future re-implementations it can be confusing if boards are discontinued (this effect can be observed for example with the OpenSpinMicroscopy: an open-source integrated microscopy platform, by Emilio J Gualda and colleagues).

Some minor textual suggestions:

The paragraph from the lines 101 to 106 might be better suited for the discussion.

169 artifacts (in AE)

In line 170 removing the 'high' and the 'image' might render the sentence more readable. Also, it is a bit long.

In line 209, taking instead of to take.

235 ..to scan quickly..

240 ..to visualize features of interest best..

524 ..the complexity of the system..

Line 548 the sentence is corrupted.

Reviewer #2 (Remarks to the Author):

Markwirth et al present live 2D SIM reconstruction using GPU based post-processing. This allows the user to see a SIM reconstruction 'live' with very little time delay. This is a very useful development and I can see great potential for SIM users to get better data. Indeed, SIM has traditionally been a two step process, first one acquires the raw data and which is then reconstructed later. This has the risk that a lot of useless data is produced, as some critical optical parameters are not always correctly adjusted. Seeing a live reconstruction allows user interaction

in real time to improve the final SIM image.
Thus this is a very useful contribution to the SIM user community.

Some comments:

-Many images are riddled with reconstruction artifact, mainly in the background: this is not a good advertisement for SIM, which has been blasted by other groups for being (too) artifact prone. To me, it seems that out-of-focus blur might be the root cause. Could the authors try to do something about that, like increasing the incidence angle of the laser beams? The authors may also refer to literature where such blur artifact in TIRF-SIM are being suppressed (Fiolka, Reto. "Clearer view for TIRF and oblique illumination microscopy." *Optics express* 24.26 (2016): 29556-29567.)

-Could the algorithm also be adapted for slice by slice 3D SIM reconstructions? 3D SIM is maybe the more widely used modality and many users could benefit from this.

Figure 3: Can the authors indicate with arrows what are defocusing artefacts and what they believe are sample structures of interest (fenestrations)? To me, all images of the red channel show some striped reconstruction artefact.

Figure 4: background appears to be clipped, please provide images that have not been background clipped. FWHM of the beads would be nice, together with a comparison to beads adherent to a coverslip. Also, 200nm beads are too big to judge if the SIM resolving power is maintained during the fast motion. I suggest using 50nm beads.

Figure 5: for a TIRF image of mitochondria, there is a lot of out-of-focus blur, which seems to cause artifacts in the reconstruction. What incident angles were used for TIRF and could it be that one or more beams were not properly in TIR?

Figure 5 What ultrastructure within the mitochondria are the authors claiming to see? Christae? Why did the authors not use the setup for ferroelectric SLM pioneered by Kner et al (polarizing beam splitter, SLM normal to optical axis)? The 4 degree tilt of the SLM will slightly change the incidence angle of each of the different illumination beams. Can the authors comment by how much? Further, how much % of the input light do they get into the first diffraction orders? What is the input polarization of the laser onto the SLM, how is it controlled, and how does it matter for this scheme?

Line 474: "An order of magnitude" would sound better to me than "a magnitude".

Suppl. Figure 7: 200nm beads are much too big to evaluate the SIM resolution. Why not using 50nm beads, which would allow an estimate of the resolution on an isolated bead?

Suppl Methods Fig 8: Why does the Wiener filtering in (b) cause so much ringing?

Overall, this is a timely manuscript and I hope the authors can address these points.

We would like to thank both reviewers for their helpful comments. We have revised the manuscript based on their suggestions and hope that they will find the revised version satisfactory and ready for publication.

In the following, please find a point-by-point response to all of the comments provided by the reviewers. The original comments are shown in regular font, while our response is shown in *italics*.

Reviewers' comments:

Reviewer #1 (Remarks to the Author):

In the paper Video-rate multi-color structured illumination microscopy with simultaneous real-time reconstruction by Andreas Markwirth and colleagues, the authors present a new approach, they dubbed VIGOR for open-source real-time reconstruction of SIM data using GPU-acceleration. The manuscripts aims to overcome the problem that acquisition and reconstruction are typically uncoupled in SIM imaging. The authors implement and adapt the earlier published fastSIM approach and combine it with a GPU accelerated network version of their fairSIM solution. The authors demonstrate their approach on mitochondria dynamics, beads, and fenestrations in LSECs. The article is timely and provides the reader with a blueprint to build a fast SIM system that provides an immediate result. This is an highly desired improvement for the workflow of SIM imaging. While the article is well and clear written, some questions and remarks remain and require a major revision.

The article is making an argument about using real-time reconstruction for super-resolution imaging. The authors should refer to the SRRF article (Gustafsson N, et al (2016). Fast live-cell conventional fluorophore nanoscopy with ImageJ through super-resolution radial fluctuations. Nature Communications, 7. doi:10.1038/ncomms12471) as it offers super-resolution and has been implemented on ANDOR cameras directly for real-time processing.

It is understood that the authors not necessarily have an ANDOR SRRF setup or access to FPGAs or related chips on the camera to implement a similar scheme and it is appreciated that the solution presented here is openly available. However, the article should at least be discussed. Also since there is also an open source implementation a comparison of the resolution performance can be made.

Response: We thank the reviewer for this suggestion and have now included a reference to the article by Gustafsson et al. Arguably, SRRF and SIM are complementary techniques, in the sense that SRRF derives super-resolution from stochastic fluctuations of specific fluorophores under constant illumination, while SIM relies on stable fluorophores under modulated excitation. The work on on-the-fly processing of SRRF data demonstrates that such techniques providing immediate super-resolution images are indeed welcomed by the microscopy community.

The authors describe the possibility to use their implementation for life cell imaging and demonstrate this with biological samples. However, one concern is the photo-toxicity. Here the effective light dose should be discussed. It could be helpful for the reader to put the results it in perspective to other techniques like SOFI (Super-

resolution optical fluctuation imaging, Dertinger and colleagues) related techniques; and SRRF, which promotes its applicability for live cell imaging and using conventional dyes like GFP. The advantages and disadvantages should be discussed. Again a direct comparison of data would be preferred.

Response: we thank the reviewer for this helpful suggestion. We have now included a comparison and discussion of the different imaging conditions required for SIM, SOFI and SRRF.

Text added to the discussion: "Other complementing super-resolution techniques include those based not on excitation light modulation, but on analyzing the stochastic fluctuation or blinking of fluorescent probes. Both, single-molecule localization techniques, which rely on very high fluorophore sparseness [32, 33] and stochastic techniques, which are able to cope with much higher emitter densities [34, 35] already feature implementations of accelerated, on-the-fly data processing, for example [36] for Single Molecule Localization Microscopy (SMLM) and [37, 38] for Super-Resolution Radial Fluctuations (SRRF) microscopy. Arguably, these techniques are somewhat orthogonal to SIM: They require much less complexity in imaging instrumentation, with data acquisition typically possible on a high-end wide-field microscope equipped with laser light sources and a fast, sensitive camera. However, they require specific fluorophores and sample conditions to work well, and typically need to acquire some hundreds (SOFI, SRRF) to thousands (dSTORM) of images for a super-resolved result."

In the article, the authors uses nine images for the reconstruction as compared to the higher precision 25 image reconstruction for obvious timing reasons. However, can a schematic for 25 images be deployed? There could be applications of slower dynamics combined with the need for more precision/resolution. How would the timing be affected?

Response: The approach of using 9 images for SIM reconstruction is the most conventional application for achieving decent results for 2D-SIM implementations. 3D-SIM with both lateral and axial resolution improvement requires the acquisition of 15 images per plane (or 25 if 5 angles are used), and only nonlinear SIM approaches utilizing photoswitchable fluorophores, such as the recently demonstrated fast NL-SIM implementation by Li et al. require the acquisition of even more images per plane. More images could be used to improve 2D-SIM, but the improvement is marginal. Other implementations have used down to 4 images (for 2D samples) without loss of information, but these methods are typically less robust and may also bleach patterns into the sample. We have added a brief explanation in the text that discuss such alternative possibilities. Our microscope is currently optimized for 9 image acquisition, but the software also works for other implementations and has been tested with data from other systems, i.e. the OMX from GE Healthcare and the Zeiss Elyra.

Text added to the Results-section: "For each color channel, the system acquires 9 raw SIM images, consisting of 3 phases and 3 rotations of the SIM pattern per time point. This scheme allows for a direct mathematical reconstruction of the super-resolved image [1, 3, 22], which can be achieved by a robust and efficient reconstruction algorithm. Standard background reduction algorithms for 2D SIM can be applied on the fly [16, 17] and all raw data is stored to still allow for more advanced and especially computationally expensive post-processing [23, 24]."

Along the same lines but for speeding up the acquisition, Meinel and colleagues, discuss schemes for REDUCING DATA ACQUISITION FOR FAST STRUCTURED ILLUMINATION MICROSCOPY USING COMPRESSED SENSING. Could these be used for the current implementation and could the imaging speed be still increased? Will these be implemented?

Response: Compressed sensing makes sparsity assumptions, which makes such methods very dependent on the actual target of the investigation. As discussed above, SIM can, even without such assumptions, be reduced to 4 images per reconstruction [22]. However, we opted against this in favor of increased robustness and fast reconstruction. We utilize 9 images because this creates a well-defined equation system, which can be solved directly without resorting to iterative approximation. Beyond this, it also allows to infer linearity properties of the resulting SIM signal, that are harder to prove or assume for deconvolution-like approaches.

One of the biggest advantages of SIM is the ease to use multiple colors painlessly as compared to other super-resolution techniques. The article claims video-rate multi-color SIM imaging. In Figure 5 the use for live cell imaging using mitotracker is demonstrated. This figure is a missed opportunity and the potential to reconstruct a biological sample live in 3 colors should be demonstrated. Also proving and actually measuring dynamics is crucial here. Potential examples for applications would be focal adhesion points or the tracking of membrane receptors. While other options demonstrating the potential would be perfectly fine, the imaging of cell surface clusters offers the opportunity to compare the current approach and its potential with other higher resolution methods, like single molecule techniques, which are typically slower. What are the limits of the setup with respect to molecular quantities; can single molecules be imaged? Can single GFP molecules be visualized? In this context, it would be relevant for the reader to know to what extent the intensity values can be interpreted and be used for measurements.

Response:

Concerning 3-color live cell imaging:

We have updated the experimental section with live-cell datasets obtained at high frame rates with two-color imaging. The instrument hardware can provide even higher framerates and 3 color imaging (which is demonstrated on e.g. the synthetic bead samples). However, the biological samples currently available to us are limiting in terms of photo-damage and availability of stable dyes for high-speed three-color imaging. This is a common observation, with current generation instrument speeds overtaking the development of photostable dyes and noise-free reconstruction algorithms. In general, our development was focused on providing the instrumentation, biological studies and work focusing on the issue of photostability is to follow.

Concerning single-molecule imaging in SIM:

The SIM method is not specifically tailored to imaging single molecules. Yet its sensitivity is sufficient to do so, if the molecules are not blinking (with the latter being problematic with typical SMLM probes as eGFP). We added a few lines to the supplementals to reflect these considerations:

“The sensitivity of current-generation sCMOS cameras, given by their quantum efficiency and read-noise, is very high, easily reaching levels suitable for single-

molecule detection (although single fluorescent molecules typically lack the linear behavior required for SIM due to blinking and bleaching effects)."

Concerning the intensity values obtained through SIM:

*We use the 'classical' direct reconstruction approach as introduced by Gustaffson and Heintzmann. The data processing involved (frequency decomposition, shifts and OTF compensation) is fairly linear, i.e. there is no iterative deconvolution process involved that typically leads to non-linear intensity behavior. However, absolute photon counts are typically **not** maintained by the reconstruction procedure.*

It is appreciated that the authors time stamp the images and that additional metadata can be added. It would be desirable to automatically add the reconstruction parameters to distinguish different version later easily and likewise being compliant with standard OME (Open Microscopy Environment) formats for the metadata so that specific images can be searched on repositories.

Response: Reconstruction parameters are already being stored as part of the imaging parameters, OME is a larger scale effort, which we are pursuing with partners (e.g. Micron Oxford) as a larger community effort. As of now, data can be converted to standard TIFF files for compatible storage, and reconstruction parameters are saved in human-readable XML files compatible with the offline fairSIM plugin. Acquired data can also be reprocessed using the GPU-accelerated live-SIM software.

One of the problems that haunt the reconstruction of SIM in my view is that the algorithm uses a noise estimate that depending on the sample can be off and needs to be adapted during the reconstruction. This holds true especially for dimmer samples, for reference see (Superresolution live imaging of plant cells using structured illumination microscopy, by George Komis and colleagues). The current manuscript demonstrates the live reconstruction of relatively bright samples. The question, therefore, is how to handle dimmer biological samples and how to adapt for example noise parameters or frequency weighting on the fly.

Response: Thanks for this comment. Our current software provides the user with the ability to adjust Wiener filter parameters and attenuation strength conveniently by sliders during the operation. This enables the user to adjust the parameters to compensate for noise and contrast. In any case, the reconstructed image will be shown instantly, while during offline reconstruction denoising algorithms, etc. can be used. All image raw data and meta data used for image acquisition and reconstruction are being saved as part of the original raw data and can be accessed and used for further processing afterwards.

A cell can easily exceed the field of view described in the present manuscript; therefore the question is, can bigger areas be imaged?

Response:

The general concept of SIM is not limited to a specific field-of-view. The instrument has to ensure a non-aberrated SIM illumination and fluorescent detection over the full

desired field-of-view. Thus, the objective lens used (high NA with simultaneously aberrations-free large field of view) is typically the most limiting component. Beyond this, both the detection sensor size and, in an SLM-based SIM, the SLM chip size, have to match this field of view while maintaining a Nyquist-sampled pixel size (projected pixels 2x under the resolution limit). With the latest technology available, fields of view with 2000x1500 pixels should be possible if objective lens quality and laser powers permit.

We have added the following statement to the supplementals:

“The achievable field-of-view is dependent on the line spacing on the SLM and its resolution: The line-spacing in the sample plane is fixed by setting a desired resolution enhancement for the SIM process, and it is linked to the SLM’s line spacing through the magnification of the SLM into the sample plane. Thus, changes in magnification can be used to tweak the SLM line spacing at constant sample-plane spacings. Due to the pixelated nature of the SLM, fine line spacings will show stronger serration, leading to a decrease in diffraction efficiency and less illumination intensity, while coarser line spacings reduce the available field-of-view. The magnification M chosen for this setup is therefore a compromise between a bigger field of view with thin lines on the SLM and a smaller FOV with thicker lines on the SLM and better diffraction efficiency.

The diffraction efficiency is approximately 5% with 488 nm and higher for longer wavelengths, as the line-thickness on the SLM scales with the wavelength.

SLMs with higher resolution could be used to illuminate a larger FOV or increase the diffraction efficiency.”

Minor comments:

The authors use the camera link for their implementation. It is not clear if that is required or if the USB3 version would be sufficient, given that only 512x512 pixels are read out. Also what bit depth is used?

Response:

In principle, modern USB 3 and 3.1 connections should provide enough speed for this specific (512x512 frames) applications, but will be saturated when using larger fields-of-view in future applications. Thus, we opted to for the Camera-Link version (a choice that for PCO has to be made at purchase, while other manufacturers allow for upgrades). Also, new generation back-side illuminated sCMOS chips have again changed timing characteristic, which also now become widely different between manufactures.

We have added a corresponding discussion to the supplementals:

“In both cases, the Camera Link variants of the cameras were used with 16 bit depth and the fast readout mode. Other cameras and interface types (e.g. current-generation sCMOS cameras with USB 3.0 and 3.1 interfaces) should – in principle – also be capable of providing similar speeds to those reached by our system. However, an in-depth understanding of the camera’s timing characteristics as well as interface and firmware capabilities might be necessary to ensure compatibility.”

The authors use three computers for the three cameras. As they also implemented the Hamamatsu cameras and the HOKAWO software from Hamamatsu supports the parallel readout of multiple cameras on one computer. Could that be implemented similarly for VIGOR? What are the bottlenecks?

Response: We have also used VIGOR in a mixed environment (using PCO and Hamamatsu cameras at the same time).

It is, in principle, possible to run both data acquisition from all cameras and the reconstruction process on one machine. However, running with 3 color channels at high frame rates, we have noticed that the high system load leads to random frame dropouts and thus data loss. This is probably due to a driver-issue with the PCO Edge 4.2 cameras with the "Camera-Link" Interface. Other cameras, for example the "Camera Link High Speed"-version with its 4-channel capture card might not have this limitation. Distributing the camera readout and post-processing onto multiple machines alleviates this problem altogether. Thus, on a system not pushed for highest frame-rates, a single computer should suffice, while for highest performance and increased flexibility the networking feature has been implemented

The authors use three cameras and if I understand correctly only read out 512x512 pixels. However, the chip size of the used cameras is 2048X2048 pixels. Somehow this raises the question if an 'old fashioned' image splitter that projects different wavelengths on the same chip can be used?

Response: This is an excellent question and the answer is "in principle, yes". However, it turns out that, as we briefly discussed in the text, the use of an image splitter for simultaneous, multi-channel acquisition with a single camera, results in an overall slower implementation of the system, as the interleaving of camera readout (one color channel) with simultaneous exposure (other color channel) relies on independent readout timing of both cameras. Additionally, a high-quality image spitting device easily reaches the price range of current sCMOS cameras, so the loss in flexibility does not save on investments.

We have expanded this paragraph in the paper to make this point more prominent: "The multi-camera approach has been chosen over an approach with only one camera and a beam-splitter, as this enables the concurrent exposure of the sample and camera-readout, which results in a speed-up that is required to match the timing-constraints set by the FLCoS (see Suppl. Note 1 for details)."

Could the timing of the cameras be used directly? It is understood that the timing depends on the SLM, but since the used cameras offer intricate trigger possibilities, it is intriguing to use a simpler glue logic, than an Arduino board. While the Arduino board is inexpensive, for future re-implementations it can be confusing if boards are discontinued (this effect can be observed for example with the OpenSpinMicroscopy: an open-source integrated microscopy platform, by Emilio J Gualda and colleagues).

Response: This is an excellent suggestion, indeed, and yes, the system is mostly independent of the board that is used to synchronize all components. The main caveat that currently limits the system performance and the timing is based on the ferroelectric SLM, which requires specific image sequences and also limits their timing. Alternative light modulators or illumination approaches will overcome this limitation.

The Arduino platform is just a convenient way to use different microcontrollers. The controller's task is fairly simple and alternative solutions could easily be created with

different types of microcontrollers ("Arduino" or others (AVR, PIC, ...)), FPGAs or computer-controlled I/O-devices.

Since the controller can run arbitrary programs and acts as a master controlling the SLM and each camera separately, it makes it easier to combine cameras of different types with different readout-modes, trigger-modes and I/O-capabilities than to let the SLM and the cameras interact with each other, making this approach more flexible in this regard.

Some minor textual suggestions:

The paragraph from the lines 101 to 106 might be better suited for the discussion.

169 artifacts (in AE)

In line 170 removing the 'high' and the 'image' might render the sentence more readable. Also, it is a bit long.

In line 209, taking instead of to take.

235 ..to scan quickly..

240 ..to visualize features of interest best..

524 ..the complexity of the system..

Line 548 the sentence is corrupted.

Response: These are all greatly appreciated comments and we have implemented them in the revised version of our manuscript.

Reviewer #2 (Remarks to the Author):

Markwirth et al present live 2D SIM reconstruction using GPU based post-processing. This allows the user to see a SIM reconstruction 'live' with very little time delay. This is a very useful development and I can see great potential for SIM users to get better data. Indeed, SIM has traditionally been a two step process, first one acquires the raw data and which is then reconstructed later. This has the risk that a lot of useless data is produced, as some critical optical parameters are not always correctly adjusted. Seeing a live reconstruction allows user interaction in real time to improve the final SIM image.

Thus this is a very useful contribution to the SIM user community.

Some comments:

-Many images are riddled with reconstruction artifact, mainly in the background: this is not a good advertisement for SIM, which has been blasted by other groups for being (too) artifact prone. To me, it seems that out-of-focus blur might be the root cause. Could the authors try to do something about that, like increasing the incidence angle of the laser beams? The authors may also refer to literature where such blur artifact in TIRF-SIM are being suppressed (Fiolka, Reto. "Clearer view for TIRF and oblique illumination microscopy." Optics express 24.26 (2016): 29556-29567.)

Response: We thank the reviewer for this comment and appreciate the suggestions. We have included the reference as suggested and propose this as a useful approach to reducing artifacts in the revised manuscript. We have also reprocessed some of the data to further reduce artifacts and discuss this and other means in the revised

document.

-Could the algorithm also be adapted for slice by slice 3D SIM reconstructions? 3D SIM is maybe the more widely used modality and many users could benefit from this.

Response: This is indeed possible, and the updated version of the fairSIM code already allows for slice by slice reconstructions. The implementation into a fast, live 3D-SIM is planned for the near future and another paper.

Figure 3: Can the authors indicate with arrows what are defocusing artefacts and what they believe are sample structures of interest (fenestrations)? To me, all images of the red channel show some striped reconstruction artefact.

Response: We have decided to replace figure 3 in the main text with one showing U2OS instead of LSECs. This dataset was acquired at an early stage of the setup-development and, in our opinion, does not represent the current state of the setup anymore. We have replaced figure 3 (which showed LSECs) with a more recent measurement showing mitochondria and ER in U2OS cells. Subsequently, we have also removed the LSEC-figures (figures 8 and 9) from the supplementals.

Figure 4: background appears to be clipped, please provide images that have not been background clipped. FWHM of the beads would be nice, together with a comparison to beads adherent to a coverslip. Also, 200nm beads are too big to judge if the SIM resolving power is maintained during the fast motion. I suggest using 50nm beads.

Response: We have updated the figure to include the background. We have also replaced figure 7 in the supplemental information (200 nm TetraSpeck beads) with a figure showing 40 nm FluoSpheres Microspheres. The 40 nm beads are, however, unfortunately not bright enough to also use them for imaging diffusing beads.

Figure 5: for a TIRF image of mitochondria, there is a lot of out-of-focus blur, which seems to cause artifacts in the reconstruction. What incident angles were used for TIRF and could it be that one or more beams were not properly in TIR?
Figure 5 What ultrastructure within the mitochondria are the authors claiming to see? Cristae?

Response: We apologize if we confused the reviewer, but no TIR illumination was used in any of the images of this manuscript. All data are 2D-SIM images. The out-of-focus blur can only be controlled by OTF attenuation. Also, we have revised the manuscript to indicate that cristae, or groups of cristae, are visible in Fig. 5.

Why did the authors not use the setup for ferroelectric SLM pioneered by Kner et al (polarizing beam splitter, SLM normal to optical axis)? The 4 degree tilt of the SLM will slightly change the incidence angle of each of the different illumination beams. Can the authors comment by how much? Further, how much % of the input light do they get into the first diffraction orders?

What is the input polarization of the laser onto the SLM, how is it controlled, and how does it matter for this scheme?

Response: We appreciate the suggestion of the reviewer and explain our reasoning in detail below:

Regarding the angled illumination:

Traditionally there are two ways of illuminating ferroelectric liquid crystal modulators. Some [7] use a polarizing beam splitter and a half waveplate to warrant orthogonality and others [19] sacrifice strict orthogonality for having fewer optical components.

The SLM is rotated by 2 degrees around the vertical axis, so the pattern on the SLM appears to be compressed in the horizontal direction.

A 2 degree tilt will effectively "shrink" the spacing by a factor of $\cos(2^\circ) = \sim 0.9994$, which is insignificant and lower than the usually used maximum allowed deviation of the pattern-spacing in the pattern-search algorithm.

One could argue that the two additional optical elements (e.g. the beam passing 8 additional surfaces) introduced in the orthogonal implementation can lead to losses and possibly unwanted stray reflections. For these reasons we adopted the oblique SLM illumination strategy, but are convinced that the other strategy also has its advantages.

Regarding the polarization:

In both cases (PBS and angled illumination) the first diffraction orders only contain the "wanted" polarization and the zero order is blocked.

The SLM is installed with the flat-band cable pointing upwards, and the laser light is shone onto the SLM with vertical polarization (electrical field). This gave the most even illumination intensity for different angles of the pattern, although this seems to be wavelength-dependent.

The polarization is selected by rotating the output coupler of the polarization maintaining fiber.

Regarding the diffraction efficiency:

About 5% of the 488 nm light is transmitted through the Fourier filter mask. For longer wavelengths the diffraction efficiency increases, as the lines displayed on the SLM get thicker, which reduces the unwanted diffraction effect of the square pixels.

We have added the following text to the supplemental information: "The diffraction efficiency is approximately 5% with 488 nm and higher for longer wavelengths, as the line-thickness on the SLM scales with the wavelength.

SLMs with higher resolution could be used to illuminate a larger FOV or increase the diffraction efficiency."

Line 474: "An order of magnitude" would sound better to me than "a magnitude".

Response: We have revised the manuscript accordingly.

Suppl. Figure 7: 200nm beads are much too big to evaluate the SIM resolution. Why

not using 50nm beads, which would allow an estimate of the resolution on an isolated bead?

Response:

Typically, 200nm beads, especially illuminated at 568nm and 647nm, provide the most immediate feedback of the SIM resolution improvement, as their size close to the diffraction limits does not allow to distinguish them in wide-field, but lets them appear clearly separated in SIM. However, we agree this is not ideal for quantitative resolution measurements, and have thus added a sample with smaller (40nm) beads to the manuscript.

Suppl Methods Fig 8: Why does the wiener filtering in (b) cause so much ringing?

Response:

The original data set used for Fig. 3 was of poor quality leading to high ringing artifacts. We have decided to replace figure 3 in the main text with one showing U2OS instead of LSECs and removed figures 8 and 9. These datasets were acquired at an early stage of the setup-development and, in our opinion, do not represent the current state of the setup anymore.

Overall, this is a timely manuscript and I hope the authors can address these points.

Reviewers' comments:

Reviewer #1 (Remarks to the Author):

In the significantly improved revised manuscript called 'Video-rate multi-color structured illumination microscopy with simultaneous real-time reconstruction' by Markwirth and colleagues, the authors present a new approach, they dubbed VIGOR for open-source real-time reconstruction of SIM data using GPU-acceleration.

Their solution offers real-time reconstruction for SIM reconstructions and is timely and technically state-of-the-art.

In the revised version the authors made highly appreciated efforts to improve the manuscript and importantly the images. Also, they provide valuable discussions in the answers to the reviewers' comments.

Unfortunately, the authors argue some of the comments away and add paragraphs to the discussion, which is fine to some extent. While I would have appreciated for example a real one to one comparison of the data, I can live with the fact that SIM and SRRF and SOFI are established enough that this may not be necessary.

However, the comment on the three-color imaging feels a bit lame. If the bleaching is too high, it won't work on the two dyes demonstrated. Also, it would be a bit pointless to do the live real-time reconstruction if it can't be applied. It is understood that for a technical manuscript no high-end biological engineering is required. As a proof of concept, regular of the shelf organelle dyes like available from Thermo-Fisher would be fine, (e.g., LysoTracker Green; BodipyTR and Mitotracker DeepRed; or even a nuclear stain would do), in essence expanding on what the authors already did. For the moment the paper claims multicolor but demonstrates only dual color on realistic samples.

Overall, the manuscript is timely, and the improvements are highly appreciated. Consequently, I hope that the authors can address the live multicolor point.

Reviewer #2 (Remarks to the Author):

I have received the response by Markwirth et al. They have addressed my main concerns. While some smaller points should be addressed, I can recommend this work for publication.

Maybe the most important point, I did indeed not realize that this work was performed in a widefield 2D SIM mode. This likely explains some of the artifacts and I am surprised that it worked so well in that regard.

I would encourage the authors to discuss in the manuscript the issue of out-of-focus blur more clearly. I.e. saying that other SIM modalities can better suppress it and hence might have the potential to suppress some of the artifacts more robustly. In addition, 2D SIM has so far mainly been applied to thin samples (or thin optical sections, as in TIRF-SIM), so the authors work is venturing a little bit in new territory (which is a good thing).

An additional clarification might be good:

Was the SIM pattern adjusted such that there was some overlap of the sidebands with the missing cone of the central OTF? Please indicate this in the supplementary methods (e.g. what was the line spacing of the SIM pattern compared to the NA of the objective? This could give an experienced user the desired information).

Given that this was widefield 2D SIM data, a certain amount of artifacts is expected. It would be nice to see the 3D SIM slice by slice processing of fairSIM implemented into the fast processing code. Maybe this can be done in the meantime?

In my view, 3D SIM slice by slice live processing would be extremely valuable. This method can better deal with out-of-focus blur and hence if it could be done live, more complicated samples

could be imaged on the fly. I think that would be really great and potentially a game changer. Thus I encourage the authors to include it, if possible.

I appreciate that the Hessian SIM processing is shown as well, in an attempt to suppress some of the artifacts.

Lastly, given that this work was not done in TIRF SIM, my suggestion for "Fiolka, R., 2016", is not as fitting as I thought (it deals with TIRF blur artifacts, which I thought were the culprit here). While it can be cited, it is more peripheral to this work.

I appreciate the other changes the authors made upon my requests, which in my view make the manuscript clearer and stronger.

Sincerely,
Reto Fiolka

We would like to thank both reviewers, again, for their helpful comments. We have now revised the manuscript once more based on their fruitful suggestions and hope that now both reviewers will find the revised version satisfactory and ready for publication.

In the following, please find a point-by-point response to all of the comments provided by the reviewers. The original comments are shown in regular font, while our response is shown in *italics*.

Reviewers' comments:

Reviewer #1 (Remarks to the Author):

In the significantly improved revised manuscript called 'Video-rate multi-color structured illumination microscopy with simultaneous real-time reconstruction' by Markwirth and colleagues, the authors present a new approach, they dubbed VIGOR for open-source real-time reconstruction of SIM data using GPU-acceleration.

Their solution offers real-time reconstruction for SIM reconstructions and is timely and technically state-of-the-art.

In the revised version the authors made highly appreciated efforts to improve the manuscript and importantly the images. Also, they provide valuable discussions in the answers to the reviewers' comments.

Unfortunately, the authors argue some of the comments away and add paragraphs to the discussion, which is fine to some extent. While I would have appreciated for example a real one to one comparison of the data, I can live with the fact that SIM and SRRF and SOFI are established enough that this may not be necessary.

However, the comment on the three-color imaging feels a bit lame. If the bleaching is too high, it won't work on the two dyes demonstrated. Also, it would be a bit pointless to do the live real-time reconstruction if it can't be applied. It is understood that for a technical manuscript no high-end biological engineering is required. As a proof of concept, regular of the shelf organelle dyes like available from Thermo-Fisher would be fine, (e.g., LysoTracker Green; BodipyTR and Mitotracker DeepRed; or even a nuclear stain would do), in essence expanding on what the authors already did. For the moment the paper claims multicolor but demonstrates only dual color on realistic samples.

In response to this comment, we retook data utilizing 3-color labeled living cells. We added 3-color images of U2OS-cells to the manuscript and attached a supplemental video. The following text was added to the manuscript: "In addition, a second live-cell experiment was carried out to demonstrate 3-color imaging of living cells. U2OS cells were stained with SYTO 9, MitoTracker Red and Tubulin Tracker Deep Red (Thermo Fisher Scientific). This experiment was conducted in a time-lapsed mode to minimize photobleaching using 10 ms illumination time per raw frame and a frame rate of 2.4 SIM-frames per second. The results are shown in Fig. 6 and Suppl. Video 3."

The figure has the following caption: "Fig. 6: Demonstration of 3-color live cell imaging with 10 ms raw frame exposure time at 2.4 (time lapsed) multi color SIM-frames per second: Live U2OS cells were stained for nucleic acids with SYTO 9, mitochondria and endoplasmic reticulum with MitoTracker Red and polymerized tubulin filaments with Tubulin Tracker Deep Red. The individual channels and an overlay of all three channels (SYTO 9 in cyan, MitoTracker Red in green and Tubulin Tracker Deep Red in magenta) are shown as reconstructed Wiener filtered wide-field and FairSIM images respectively. A scale bar in the overlay images represents 5 μ m."

The sample preparation for this experiment is further described in the corresponding section of the manuscript: "For Fig. 6 U2OS cells were simultaneously stained with three fluorophores according to the manufacturer's instructions. Cells were incubated with SYTO 9 at 2 μ M for 30 minutes (S34854, Thermo Fisher Scientific) in order to stain predominantly chromatic DNA with a distinct background signal in the cytoplasm as it has low affinity for RNA as well. MitoTracker Red CMXRos (M7512, Thermo Fisher Scientific) at a 500nM concentration for 30 minutes stained the mitochondria as well as the endoplasmic reticulum. Finally Tubulin Tracker Deep Red (T34077, Thermo Fisher Scientific) stained the polymerized tubulin. Cells were washed three times with PBS after incubation with the respective staining solution and imaged in CO₂ independent growth media (18045054, Gibco, Thermo Fisher Scientific)."

Overall, the manuscript is timely, and the improvements are highly appreciated. Consequently, I hope that the authors can address the live multicolor point.

We thank the reviewer for this feedback and hope to have satisfied her/his request with this addition.

Reviewer #2 (Remarks to the Author):

I have received the response by Markwirth et al. They have addressed my main concerns. While some smaller points should be addressed, I can recommend this work for publication.

Maybe the most important point, I did indeed not realize that this work was performed in a widefield 2D SIM mode. This likely explains some of the artifacts and I am surprised that it worked so well in that regard.

I would encourage the authors to discuss in the manuscript the issue of out-of-focus blur more clearly. I.e. saying that other SIM modalities can better suppress it and hence might have the potential to suppress some of the artifacts more robustly. In addition, 2D SIM has so far mainly been applied to thin samples (or thin optical sections, as in TIRF-SIM), so the authors work is venturing a little bit in new territory (which is a good thing).

An additional clarification might be good:

Was the SIM pattern adjusted such that there was some overlap of the sidebands with the missing cone of the central OTF? Please indicate this in the supplementary methods (e.g. what was the line spacing of the SIM pattern compared to the NA of the objective? This could give an experienced user the desired information).

The following paragraph was added to the manuscript: "The SIM imaging modality was optimized for speed (3 angles, 3 phases) and the pattern was chosen to give a lateral resolution improvement of 1.8 (at an NA of 1.33). This leaves some overlap of the side bands and thus allows for some out-of-focus reduction by reweighting the contribution of the different bands [16]. Two trade-offs are possible for optimization, both of which only require minimal changes to the microscope (replacing the order-selection-aperture and realigning the polarization filter): A coarser pattern set could be used to increase the overlap. This would improve the background suppression without the need to record more images but reduce lateral resolution improvement. Alternatively, a coarse and a fine pattern could be used consecutively. This would require additional images to be acquired and would thus slow down the instrument but allow for background reduction and full lateral resolution. Of course, 3-beam SIM is preferable, here, but this also comes with higher effort for polarization control and top-phase alignment (matching the axial pattern repetition to the detection focal plane)."

Given that this was widefield 2D SIM data, a certain amount of artifacts is expected. It would be nice to see the 3D SIM slice by slice processing of fairSIM implemented into the fast processing code. Maybe this can be done in the meantime?

In my view, 3D SIM slice by slice live processing would be extremely valuable. This method can better deal with out-of-focus blur and hence if it could be done live, more complicated samples could be imaged on the fly. I think that would be really great and potentially a game changer. Thus I encourage the authors to include it, if possible.

The live reconstruction mode offers the same capabilities as the fairSIM ImageJ / FIJI plugin, including slice-by-slice 3-beam SIM reconstruction. Indeed, we have tested the code by processing single-slice 3-beam time-lapse SIM data acquired on an OMX v4 system. If the manufacturer of such a system would implement an interface (a live network stream of the data), the live reconstruction part of fairSIM should seamlessly work on such a system

I appreciate that the Hessian SIM processing is shown as well, in an attempt to suppress some of the artifacts.

Lastly, given that this work was not done in TIRF SIM, my suggestion for "Fiolka, R., 2016", is not as fitting as I thought (it deals with TIRF blur artifacts, which I thought were the culprit here). While it can be cited, it is more peripheral to this work.

We decided to keep this reference nonetheless, as it is a useful reference for others and our own future work on TIRF-SIM.

I appreciate the other changes the authors made upon my requests, which in my view make the manuscript clearer and stronger.

Sincerely,

Reto Fiolka